# Dynamic Service Function Chain Deployment in NFV-enabled Offshore Edge Computing Networks

Yongchun Han, Bin Lin, Chaoyue Zhang, Qiaodan Wang

Information Science and Technology College, Dalian Maritime University, Dalian 116026, China

Corresponding Author: Bin Lin Email: binlin@dlmu.edu.cn

*Abstract*—**Network function virtualization (NFV) has gained significant attention as an important paradigm in network service provisioning. In an NFV environment, network services are provided flexibly by deploying service function chains (SFCs) dynamically, which consist of multiple virtual network functions (VNFs) arranged in predefined order. In this paper, we formulate the dynamic SFC deployment optimization (DSDO) problem in the NFV-enabled offshore edge computing network, by jointly optimizing VNF deployment and routing to maximize the service acceptance rate. An SFC deployment scheme, named betweenness centrality and resource availability based dynamic SFC deployment (BR-DSD) scheme is proposed to solve the optimization problem. The proposed BR-DSD scheme aims to allocate resources for SFC deployment by considering both network resource availability and betweenness centrality. Simulation results demonstrate that the proposed BR-DSD scheme outperforms baseline schemes in terms of service acceptance rate.**

*Keywords—offshore edge computing, service function chain (SFC) deployment, network function virtualization (NFV)*

## I. INTRODUCTION

The rapid growth of offshore activities, such as fishing, shipping, tourism, has driven a surge in demand for advanced maritime applications. These applications, equipped with capabilities including image processing, visual enhancement, automatic navigation, and other supporting tools, aim to provide diverse and personalized network services for offshore users. However, these applications require substantial computation resources, posing a challenge for offshore users with limited computation capacity. To address this challenge, edge computing has emerged as a promising solution, offering ample computation resources in close proximity to offshore users, which enables efficient execution of computation tasks on edge servers.

Conventional edge computing networks are typically customized for specific missions, which limits their adaptability to diverse services. To addresses this, Network function virtualization (NFV) emerges as a crucial paradigm in network service provisioning. NFV decouples network functions from dedicated hardware and virtualizes them into software components, known as virtual network functions (VNFs). This allows a single physical network to support multiple services simultaneously, enhancing service delivery flexibility and network resource utilization. Within the NFV environment, services are provided dynamically via service function chains (SFCs), which consist of multiple VNFs arranged in a predefined order. However, deploying SFCs in edge networks faces a significant challenge: limited network resources.

The SFC deployment problem has drawn significant attention from researchers. The authors in [1] separate the flows into different kinds based on resource preferences, and define relative cost to balance the resource consumption and route heterogeneous traffic at flow level differentially in SDN and NFV-enabled network aiming to minimize the resource consumption costs of flows with SFC requests. The authors in [2] propose a VNF resource allocation scheme based on context-aware grouping technology that enables groups (based on the geographic context of users, such as location and velocity) to compute the optimal number of clusters to minimize the end-to-end delay of network services. The authors in [3] accurately measure resource consumption on edge devices considering both CPU consumption by computing and by communicating between consecutive network functions in a chain on edge devices. The authors in [4] consider sharing the allocated computing capacity of a host by VNF resizing and priority queuing to minimize the service deployment cost. In [5], the subchain-aware NFV service placement optimization problem is investigated that accounts for the configuration cost for stitching together reused network functions to an SFC and strives to reuse existing subchains of consecutive network functions (with already deployed SFC traffic steering).

While above studies primarily focus on efficiently allocating network resources for deploying SFCs, they overlook the significance of key links and nodes. In reality, these key components play crucial intermediary roles within networks, naturally experiencing higher resource consumption rates. Congestion at these points can severely hinder service acceptance rates. Therefore, considering the impact of key links and nodes is crucial for improving the service acceptance rate. It is worth mentioning that the authors in [6] determine the VNF placement using a metric based on betweenness centrality and server failure rate, but without considering network resources. Furthermore, offshore edge computing networks suffer from even more severe resource scarcity compared to land-based edge computing, especially in terms of bandwidth resources, due to the changeable maritime environment. Therefore, when deploying SFCs in offshore edge networks, it is imperative to consider bandwidth resources, as opposed to some literatures that assume ample bandwidth resources by assuming optical cable connections between edge servers.

Network service requests can be classified into offline and online requests. Offline requests are provided beforehand, while

online requests are assumed to arrive sequentially without prior knowledge of future requests [7]. In edge networks, requests frequently join and leave due to user mobility. Therefore, the dynamic nature of network services in the edge networks must be taken into consideration.

In this paper, we investigate the dynamic SFC deployment optimization (DSDO) problem in an NFV-enabled offshore edge computing network, by jointly optimizing VNF deployment and routing to maximize the service acceptance rate. Given the NP-hard nature of this problem, we propose a betweenness centrality and resource availability based dynamic SFC deployment (BR-DSD) scheme to solve the optimization problem. The proposed BR-DSD scheme aims to allocate resources for SFC deployment by considering both network resource availability and betweenness centrality, mitigating the potential adverse impact of current services on subsequent ones. Our key contributions include:

- We propose the DSDO problem in the offshore edge computing network, by jointly optimizing the VNF deployment and routing aiming at maximizing the service acceptance rate.

- Due to the NP-hard, we propose the BR-DSD scheme to solve it. To enhance the service acceptance rate, we consider both the network resources and network betweenness centrality to reduce the potential adverse impact of the current service on subsequent ones in BR-DSD scheme. Simulation results show that BR-DSD scheme can efficiently solve the DSDO problem and obtain high performance in terms of the service acceptance rate.

## II. SYSTEM MODEL AND PROBLEM FORMULATION

In this section, we model the system, the network and network services in detail. Then we elaborate the constraints on SFC deployment and formulate the DSDO problem.

### A. System Model

In this paper, we consider an offshore edge computing system enabled by NFV, which consists of a High Altitude Platform (HAP) equipped with a Software-Defined Networking controller, Intelligent Buoys (IBs), and intelligent ships (ISs), as illustrated in Fig. 1. The HAP serves as a control center with computation and communication capabilities, in order to sense network state information, orchestrate VNFs, and make SFC deployment decisions. Each IB possesses computation and communication capabilities providing resources for SFCs.

In the offshore edge computing system, the SFC deployment process includes three steps after the HAP receiving a service request (SR). First, the HAP determines the optimal source node and destination node based on locations of ISs, the service types and network conditions. Second, it constructs the SFC tailored to the SR. Finally, the HAP formulates the SFC deployment scheme, that is, VNFs placement scheme and the routing scheme, providing efficient and effective service delivery.

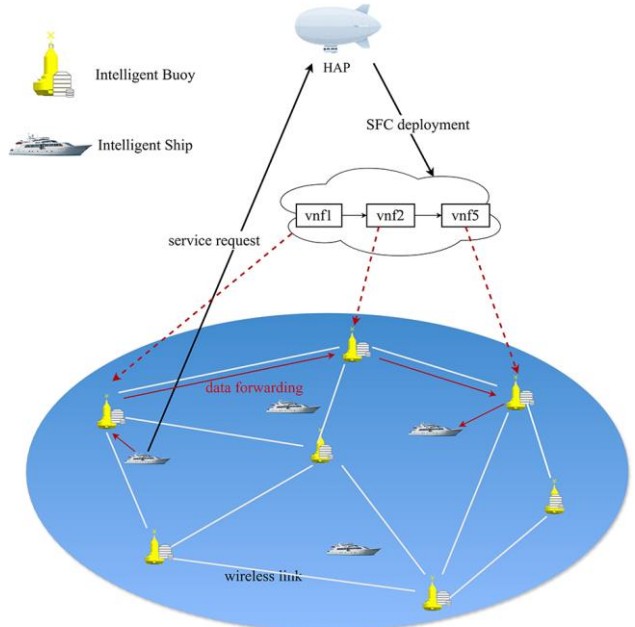

Fig. 1. NFV-enable offshore edge computing system

### B. Network Model

The offshore edge computing network is modeled as an undirected graph $G = (V, E)$, where $V = \{v\}$ represents the set of IBs responsible for instantiating VNFs and forwarding data, $E = \{e\}$ comprises

communication links interconnecting neighboring IBs. Each link corresponds to a pair of IBs, i.e., $e = (u, v), u, v \in V$ and $u \neq v$ with a total bandwidth capacity $B_e$. Each IB $v \in V$ possesses a total computing capacity $C_v$.

We assume that the hardware resources of each IB have been virtualized using lightweight container-based technology [8], enabling flexible allocation and sharing of resources. This virtualization approach facilitates the creation of VNFs with arbitrary capacity on demand, owing to the fine-grained resource allocation capabilities of containers.

### C. Service Model

To accommodate the dynamic nature of network states and SRs, we discretize a continuous period into time slots of equal length, denoted as $T = \{t_1, t_2, \cdots, t_{|T|}\}$. Assume that SRs arrive and depart randomly but processed in a first-come-first-serve manner at the start of each time slot, whose set is defined as $R = \{r\}$. $|R|$ represents the total number of SRs in a given period. At the conclusion of each time slot, the HAP checks all alive services to identify those have reached the end of lifetime. For completed services, the corresponding SFCs are terminated and the network resources are released for subsequent services.

Let $F = \{f_1, f_2, \cdots, f_{|F|}\}$ be the collection of all types of VNFs supported by the system. Each SR is represented by

$r = \{s_r, d_r, S_r, b_r, t_r^{\text{arrive}}, t_r^{\text{complete}}\}$. $s_r$ and $d_r$ represent the source node and destination node. $P_r$ is the set of possible simple paths from the source node $s_r$ to the destination node $d_r$. $S_r = \langle f_{r,1}, f_{r,2}, \cdots, f_{r,|S_r|} \rangle$ is an ordered list of VNFs comprising the SFC of SR $r$, where $|S_r|$ indicates the length of the SFC and $f_{r,i}$ represents the $i$th VNF in the chain. $b_r$ denotes the bandwidth requirement ensuring the necessary quality of service. $t_r^{\text{arrive}}$ and $t_r^{\text{complete}}$ represent the arrival and expected completion time of SR $r$.

### D. Problem Formulation

Next, we define the DSDO problem in detail. To this end, we introduce two binary decision variables $x_v^{f_{r,i}}$ and $y_p^r$. The binary variable $x_v^{f_{r,i}}$ indicates whether the $i$th VNF of the service $r$ is deployed on node $v$.

$$x_v^{f_{r,i}} = \begin{cases} 1, & \text{if } i\text{-th VNF of service } r \\ & \text{ is deployed on node } v, \\ 0, & \text{otherwise}. \end{cases} \quad (1)$$

The binary variable $y_p^r$ represents whether path $p \in P_r$ is selected for routing.

$$y_p^r = \begin{cases} 1, & \text{if service request } r \text{ selects path } p \text{ for routing}, \\ 0, & \text{otherwise}. \end{cases} \quad (2)$$

Drawing from [3], for an accurate assessment of CPU consumption, it is crucial to consider not only the computation consumed CPU but also the communication between consecutive VNFs consumed CPU in an SFC. The communication consumed CPU by VNF $f \in S_r$ is

$$c_f^{\text{comm}} = p_f^{\text{in}} + p_f^{\text{out}} \quad (3)$$

where $p_f^{\text{in}}$ and $p_f^{\text{out}}$ are the ingress traffic volume and egress traffic volume of VNF $f \in S_r$, respectively. The computation consumed CPU is

$$c_f^{\text{comp}} = \alpha_f \cdot p_f^{\text{in}} \quad (4)$$

where $\alpha_f$ is the ratio of CPU consumed by computation and by communication.

Then we elaborate the constraints when deploying SFCs.

1) VNF Deployment Constraints: For SR $r \in R$, each VNF are assumed to be deployed on only one network node, i.e.,

$$\sum_{v \in V} x_v^{f_{r,i}} \leq 1, \forall r \in R, f_{r,i} \in S_r. \quad (5)$$

For VNF $f_{r,i}, f_{r,i+1} \in S_r$, if VNF $f_{r,i}$ is deployed on node $v \in V$ of path $p$ and its subsequent VNF $f_{r,i+1}$ is successfully deployed on a node $u \in V$, then node $u$ must be on path $p$ positioned

subsequent to node $v$ within the sequence of nodes consisting of path $p$. The constraint is formally denoted as

$$x_v^{f_{r,i}} \cdot \delta_p^v = 1, x_u^{f_{r,i+1}} = 1, \forall r \in R, f_{r,i} \in S_r, f_{r,i+1} \in S_r$$
$$\Rightarrow \sum_{u \in V(v,p)} x_u^{f_{r,i+1}} \cdot \delta_p^u = 1 \quad (6)$$

where $\delta_p^v$ is a binary indicator variable. If routing path $p \in P$ passes through node $v \in V$, $\delta_p^v = 1$ and 0 otherwise. $V(v,p)$ denotes a node set, where all nodes position subsequent to node $v$ within the sequence of nodes consisting of path $p$. This constraint ensures that data flows through each VNF in the prescribed sequence.

2) Routing Path Constraint: If SR $r$ selects path $p \in P_r$ for routing, then the all VNFs in SFC must be deployed sequentially on nodes composing path $p$. This constraint is mathematically described as

$$y_p^r = 1, \forall r \in R, p \in P \Rightarrow \sum_{v \in V} x_v^{f_{r,i}} \cdot \delta_p^v = 1, \forall f_{r,i} \in S_r. \quad (7)$$

3) Computation Capacity Constraint: The aggregate consumed CPU on node $v \in V$ must not exceed the total available CPU resources on the node $v$, i.e.,

$$\sum_{r \in R_{\text{alive}}(t)} \sum_{f_{r,i} \in S_r} c_{f_{r,i}}^{\text{comm}} \cdot y_p^r \cdot \delta_p^v + c_{f_{r,i}}^{\text{comp}} \cdot x_v^{f_{r,i}} \leq C_v, \forall v \in V, t \in T. \quad (8)$$

where $R_{\text{alive}}(t) = \{r \mid t_r^{\text{arrive}} \leq t \leq t_r^{\text{complete}}\}$ denotes the set of alive services within slot $t \in T$.

4) Communication Capacity Constraint: The aggregate consumed bandwidth on the link $e \in E$ must not exceed the total available bandwidth on link $e$, which is expressed as

$$\sum_{r \in R_{\text{alive}}(t)} \sum_{p \in P_r} y_p^r \cdot \phi_p^e \cdot b_r \leq B_e, \forall e \in E, t \in T. \quad (9)$$

where $\phi_p^e$ is a binary indicator variable. If link $e \in E$ belongs to routing path, $\phi_p^e = 1$ and 0 otherwise.

Due to the shared nature of resources among multiple services within the offshore edge computing network and given the scarce CPU and bandwidth resources available, it is not feasible to guarantee the successful acceptance of all SRs. We introduce a binary variable $z_r$, i.e.,

$$z_r = \begin{cases} 1, & \text{if } \sum_{f_{r,i} \in S_r} x_v^{f_{r,i}} = |S_r| \text{ and } \sum_{p \in P_r} y_p^r = 1, \forall r \in R, \\ 0, & \text{otherwise}. \end{cases} \quad (10)$$

In this paper, our objective is to maximize the dynamic service acceptance rate. Consequently, the problem can be formulated as follows:

$$\max_{\{x_v^{f_{r,i}}\}, \{y_r^p\}} \frac{\sum_{r \in R} z_r}{|R|} \quad (11)$$
$$\text{s.t.} (1)\text{-}(10)$$

## III. APPROACH FOR DSDO PROBLEM

To address the DSDO problem, we propose the BR-DSD scheme. The primary objective of this scheme is to maximize the service acceptance rate. To achieve this goal, we devise a comprehensive assessment criterion that takes into account three key factors: CPU resources, bandwidth resources, and network betweenness centrality. This criterion is employed to score a set of SFC deployment strategy with the aim of mitigating the potential impact of the current service deployment on subsequent services.

### A. BR-DSD Scheme

| BR-DSD Scheme |
|---|
| **Input**: Offshore edge computing network $G = (V, E)$, service requests $R$ |
| **Output**: SFC deployment scheme $S$ |
| 1   Initialize $R_{alive} = \varnothing, R_{accept} = \varnothing, S = \varnothing$ ; |
| 2   Calculate the shortest path matrix $\mathbf{D}$ ; |
| 3   **for** each $t \in T$ **do** |
| 4     **for** each $r \in R_{arrive}(t)$ **do** |
| 5       Remove links for $b_e < b_r$ and nodes with less CPU resources according to (3) in $G \Rightarrow$ network $H$ |
| 6       Calculate $d_{min} = max(2, \mathbf{D}(s_r, d_r))$ |
| 7       **while** $d < d_{max}$ **do** |
| 8         Execute **Algorithm 1** to get routing paths $P_r$ ; |
| 9         **if** $P == \varnothing$ **then** |
| 10          $d \leftarrow d + 1$ |
| 11          **continue** |
| 12         **end if** |
| 13         Execute **Algorithm 2** to SFC deployment scheme $s = (p, d)$ |
| 14         **if** $p \neq \varnothing$ and $d \neq \varnothing$ **then** |
| 15          Add $r$ to $R_{active}$ and $R_{accept}$ , $S = S \cup (r, p, d), z_r = |S|$ |
| 16          Update resources in $G$ |
| 17         **end if** |
| 18         $d \leftarrow d + 1$ |
| 19       **end while** |
| 20     **end for** |
| 21     **for** each $r \in R_{arrive}(t)$ **do** |
| 22       **if** $t_r^{complete} == t$ **then** |
| 23       Release resources occupied by $r$ in $G$ |
| 24       $R_{alive}(t) = R_{alive}(t) - r$ |
| 25       **end if** |
| 26     **end for** |
| 27   **end for** |
| 28   return $S$ |

At the beginning of each time slot $t \in T$, arrival SRs, i.e., $r \in R_{arrive}(t), R_{arrive}(t) = \{r \mid t_r^{arrive} = t\}$, are processed in a first-come-first-serve manner. The processing of each SR in the offshore edge computing network consists of three primary stages.

Specifically, during the preparation stage, links with bandwidth resources less than the required are pruned to ensure the quality of service. Subsequently, checks if nodes possess sufficient CPU resources according to (3). If not, these nodes are removed. Additionally, isolated nodes, nodes with no connected links, are also removed. Ultimately, a modified network $H = (V', E')$ is constructed with the same attributes as the pruned network $G$ . In the first stage, the Depth-Limited DFS-Based Path (DLDP) algorithm is utilized to identify potential routing paths form source node to destination node within the modified network $H$ . In the second stage, based on the outcomes of the first stage, the Betweenness and Resource-based VNF Deployment (BRVD) algorithm is applied to place the VNFs along the obtained routing paths. If an optimal routing path $p \in P_r$ and VNF placement strategy $d \in D_r$ are identified, the service request $r$ itself, routing path $p$ , and VNF deployment strategy $d$ are saved as a tuple $(r, p, d)$ in both the accepted service requests list $R_{accept}$ and the alive service requests list $R_{alive}$ .The network resources are then updated accordingly.

If DLDP algorithm fails to return routing paths or BRVD algorithm fails to deploy all VNFs, increases the search depth in DLDP algorithm and repeat the SFC deployment procedure until the maximum search depth. This process is iterated until all SRs $r \in R_{arrive}(t)$ are processed. At the conclusion of the time slot $t \in T$ , network resources occupied by completed services are released, and the corresponding SRs are removed from the alive SR list $R_{alive}$ .

### B. DLDP Algorithm

| Algorithm 1. DLDP Algorithm |
|---|
| **Input**: Network $H$ , source node $s_r$ , destination node $d_r$ , depth $d$ |
| **Output**: Path set $P_r$ |
| 1   Initialize stack $Stk = \varnothing$ , path set $P_r = \varnothing$ ;// $Stk = [(node,\ depth,\ path)]$ |
| 2   Push $(s_r,\ 0,\ [s_r])$ into stack |
| 3   **while** $Stk \neq \varnothing$ **do** |
| 4     $(node, depth,\ path) = pop(Stk)$ |
| 5     **if** $node == d_r$ and $depth == d$ **then** |
| 6       Add $path$ to $P_r$ |
| 7     **end if** |
| 8     **if** $node \neq d_r$ and $hops < d$ **then** |
| 9       **for** $nbr$ in $H[node].neighbors$ **do** |
| 10         Push $\left( Stk, (nbr,\ hops+1,\ path.append(nbr)) \right)$ |
| 11       **end for** |
| 12     **end if** |
| 13   **end while** |
| 14   return $P_r$ |

The DLDP algorithm is a variant of the traditional Depth-First Search (DFS) algorithm, combining the DFS strategy with a depth constraint. It is designed to find paths from source node to destination node. Similar to DFS, the DLDP algorithm explores the network to find paths from the source node to the destination node. Due to the limit on the depth of the search, the algorithm focuses solely on paths that match the specific length, effectively avoiding returning the repeated paths when the algorithm is invoked multiple times. The DLDP algorithm first create an empty stack to save the search information as a tuple $(node, depth, path)$. Then $(s_r, 0, [s_r])$ is pushed into this stack. The algorithm pops the head element in the stack and proceeds to explore the neighboring nodes, keeping track of the current path and its depth at each step. Whenever the algorithm reaches a node, it checks whether the node is the destination and whether the current depth matches the specified limit. If both conditions are satisfied, the path is added to set $P_r$. The DLDP algorithm terminates when all possible paths from the source node to the destination node within the specified depth limit are explored and recorded. At the end of algorithm, the set of paths $P_r$ are returned.

### C. BRVD Algorithm

**Algorithm 2.** BRVD Algorithm

**Input**: Path set $P_r$, service request $r$

**Output**: optimal SFC deployment scheme $s^*$

1    Initialize $s^* = \varnothing$

2    **if** $|S_r| \geq |P_r[0]| - 2$ **then**

3      **for** each path $p \in P_r$ **do**

4        calculate $S_{\text{PATH}}(p)$ according to (12)

5        Generate all possible Structured $C$ ;

6        **for** each sub-chain $c \in C$ **do**

7          $d \Leftarrow$ Deploy sub-chain on $p$ sequentially

8          **if** $d$ meets resources constraint (8) **then**

9            calculate $S_{\text{VNF}}(d)$ according to (13)

10            $s^* \leftarrow s$ with maximum $S_{\text{SFC}}(r)$ according to (14)

11          **end if**

12        **end for**

13      **end for**

14    **end if**

15    **else**

16      **for** each path $p \in P_r$ **do**

17        Generate all possible sub-paths $P^*$ based on $p$ ;

18        $d \Leftarrow$ Deploy VNFs on $p^*$ sequentially

19        **if** $d$ meets resources constraint according to (9) **then**

20          calculate $S_{\text{PATH}}(p)$ according to (12)

21          calculate $S_{\text{VNF}}(d)$ according to (13)

22          $s^* \Leftarrow s$ with maximum $S_{\text{SFC}}(r)$ according to (14)

23        **end if**

24      **end for**

25    return $s^*$

The BRVD algorithm is designed to determine the optimal routing path and VNFs deployment locations along this path. Its primary objective is to allocate resources properly and thereby mitigate the potential impact of the current service on subsequent services. The core idea behind this algorithm is to deploy VNFs on less critical but well-resourced intermediate nodes along a less critical but adequate link. This strategy is adopted due to the inherently faster resource consumption rates on key links and nodes, which make them prone to network congestion and can subsequently lead to services acceptance failure. Therefore, it is imperative to deploy SFC on less critical links and nodes that possess sufficient resources, while ensuring that the alive SFCs remain undisrupted. This strategy minimizes the potential for network congestion and thereby enhances the service acceptance rate.

Betweenness is a centrality graph metric with global significance that can be applied to both nodes and links [9]. Node betweenness is defined as the number of shortest paths that pass through a node, while link betweenness is similarly defined as the number of shortest paths that pass through a link. Betweenness reflects the positional importance of a node or link in the network. Nodes or links with high betweenness typically play more important intermediary roles as bridges in the network, connecting different parts and being crucial for network connectivity and information flow. Consequently, we use betweenness centrality to measure the importance of a node or a link. Furthermore, we design an assessment criterion to score the routing path and VNF deployment scheme by combining computing resources, bandwidth resources, and betweenness centrality. This ensures a comprehensive evaluation that aligns with the objectives of the BRVD algorithm.

We refer to [10] to calculate betweenness of link and node. The score of routing path $p$ is calculated by

$$S_{\text{PATH}}(p) = \sum_{e \in p} \omega_1 \cdot \frac{b_e - b_r}{b_{\max}} + (1 - \omega_1) \cdot \left(1 - \frac{bc(e)}{bc_{\max}(e)}\right) \quad (12)$$

where $b_{\max}$ and $bc_{\max}(e)$ denotes the maximum remaining bandwidth and the maximum edge betweenness in the network, respectively. And $\omega_2$ is a weight factor. Similarly, the score of VNFs deployment scheme $d$ is calculated by

$$S_{\text{VNF}}(d) = \sum_{v \in d} \omega_2 \cdot \frac{c_v - c_f}{c_{\max}} + (1 - \omega_2) \cdot \left(1 - \frac{bc(v)}{bc_{\max}(v)}\right) \quad (13)$$

where $c_{\max}$ and $bc_{\max}(v)$ denotes the maximum remaining CPU and the maximum node betweenness in the network, respectively. And $\omega_2$ is a weight factor that adjusts the significance of these factors. Based on the $S_{\text{PATH}}(p)$ and $S_{\text{VNF}}(d)$, we evaluate the SFC deployment scheme of SR $r$ by

$$S_{\text{SFC}}(r) = \omega_3 \cdot S_{\text{PATH}}(p) + (1 - \omega_3) \cdot S_{\text{VNF}}(d) \quad (14)$$

where $\omega_3$ is a weight factor that determines the relative

importance of the routing path and the VNFs deployment in the overall assessment.

Before specifying the BRVD algorithm, it is imperative to clarify two fundamental concepts: Structured SFC (S-SFC) and Reconstructed Routing Path (R-RP). The S-SFC is derived from the linear SFC by segmenting the chain into distinct sub-chains, while preserving the original order of VNFs within the chain. A sub-chain represents a consecutive sequence of VNFs within the chain, as depicted in Fig. 2. In the context of a routing path, nodes designated for deploying VNFs are called deployable nodes. An R-RP is a path that shares the same nodes as the original routing path, except that some deployable nodes are solely utilized for data forwarding, as exemplified in Fig. 3.

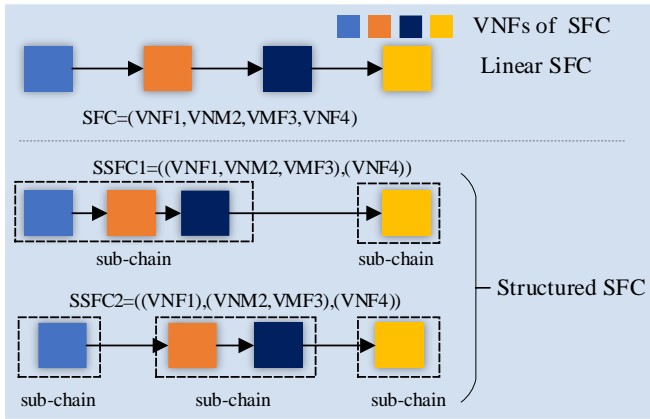

Fig. 2.  An Illustration of Structured SFC and Sub-chains

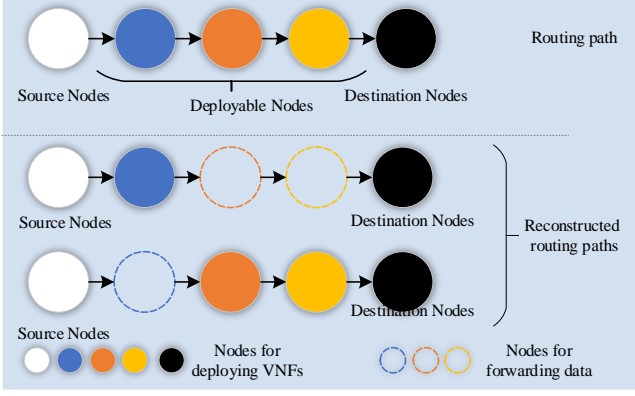

Fig. 3.  An Illustration of Reconstructed routing paths and deployable nodes

The BRVD algorithm initiates by comparing the length of the SFC, i.e., $|S_r|$, with the number of deployable nodes, i.e., $|p|-2$, in the routing path. If the length of the SFC is greater than or equal to the number of deployable nodes in the routing path, i.e., $|S_r| \geq |p|-2$, the following steps are executed. Initially, the original SFC is structured to obtain a set of S-SFCs, each of which has a length equivalent to the number of deployable nodes in the routing path. Subsequently, the sub-chains within each S-SFC are sequentially deployed onto the deployable nodes of the routing path. Furthermore, VNF deployment schemes that fail to meet the CPU resource

constraints are excluded. Ultimately, the optimal SFC deployment scheme is selected based on equation (13).

Conversely, if the length of the SFC is less than the number of deployable nodes in the routing path, the algorithm constructs routing paths with a length equal to that of the SFC. VNFs are then sequentially deployed onto these R-RPs. Similar to the previous case, VNF deployment schemes are filtered, and the optimal SFC deployment scheme is selected and returned.

## IV. PERFORMANCE EVALUATION

To evaluate the performance of the proposed algorithm in the offshore edge computing network, we developed a simulator using Python. The topology of the offshore edge computing network was generated using the Python package NetworkX 3.2.1, consisting of 10 nodes and 17 links, i.e, $|V|=10, |E|=17$. The nodes are randomly distributed in an offshore environment. An edge $e=(u,v)$ indicates a wireless communication link between node $u$ and $v$ with a bandwidth capacity of 15 megabits per second (Mbps). We assume that the bandwidth consumption remains constant as the flow traverses VNF instances, and the traffic rate along each link of the routing path does not fluctuate during the service's lifetime. Each node has a CPU capacity of 100 Mbps.

In the offshore edge computing network, 10 different types of VNFs are offered, i.e., $F=\{f_1, f_2, \cdots, f_{10}\}$. For each SR, source node and destination node are randomly chosen from distinct nodes in the offshore edge computing network, respectively and VNFs are randomly chosen from the VNF set varying from 3 to 4. The SRs follow a Poisson distribution. The lifetime of each request varies from 5 to 10 time slot. The ratio of CPU consumed by computation and communication is randomly and uniformly set between 1 and 3 [3]. The results are obtained by averaging 20 simulation runs to ensure reliability and accuracy.

We compare the performance of our proposed scheme with two baseline schemes: Resource Balanced Dynamic SFC Deployment (RB-DSD) scheme and the Resource Greedy Dynamic SFC Deployment (RG-DSD) scheme. In the RB-DSD scheme, each VNF within an SFC is deployed on one node along the shortest routing path with the aim of achieving resource balance by distributing the VNFs on multiple nodes. Conversely, in the RG-DSD scheme, all VNFs within an SFC are deployed onto a single node with the maximum CPU resources along the shortest routing path, in order to minimize bandwidth and CPU resources consumption.

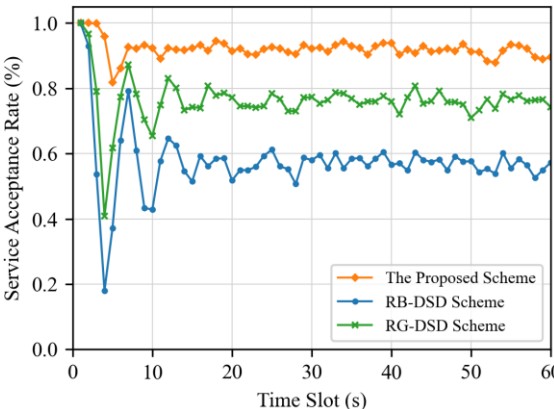

Fig. 4. The dynamic service acceptance rate of each time slot.

Fig. 4 depicts the dynamic service acceptance rate across each time slot. Across all three schemes, the service acceptance rate exhibits minor fluctuations over time while maintaining overall stability. Notably, our proposed scheme demonstrates superiority over the other two, with the most stable dynamic acceptance rate. This indicates that our scheme not only effectively enhances the service acceptance rate but also exhibits better adaptability to dynamic changes in services. Fig. 5 illustrates the relationship between the service acceptance rate and the average number of SRs in each time slot. As the number of service requests increases, the service acceptance rate gradually decreases for all three schemes. This decline can be attributed to the limited bandwidth and computation resources, which constrain the capacity to accommodate a larger number of services. Furthermore, it is evident that the proposed BR-DSD scheme outperforms the other two schemes.

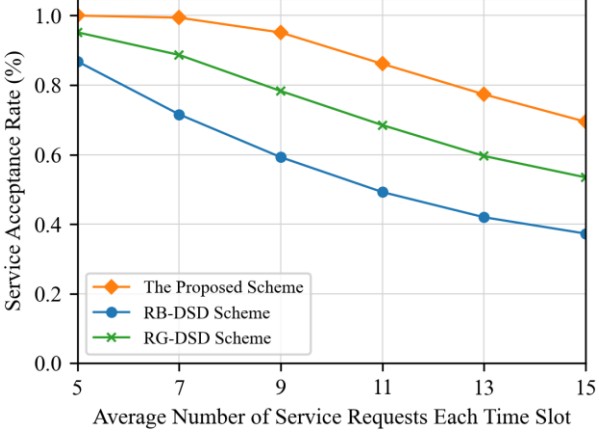

Fig. 5. The relationship between the service acceptance rate and the average number of service requests in each time.

The superiority of our proposed scheme can be attributed to its dynamic deployment of SFCs based on the remaining network resources and the network topology. Specifically, our scheme prioritizes deploying VNFs on less critical nodes with ample computation resources, along less critical routing paths with ample bandwidth resources. This strategy effectively reduces the likelihood of congestion on key links and nodes,

thereby mitigating the potential impact of the current service on future service requests and enhancing the overall service acceptance rate. In contrast, the RG-DSD scheme deploys SFCs on nodes with the maximum remaining resources along the shortest path, which reduces the immediate consumption of CPU and bandwidth resources but can easily lead to network congestion, subsequently consuming more network resources. On the other hand, the RB-DSD scheme achieves balanced resource consumption but does so at the cost of consuming more bandwidth resources and computation resources for communication.

Fig. 6 depicts the relationship between the average acceptance rate and the length of SFC. As the length of SFC increases, the network experiences increased resource consumption, resulting in a decline in the service acceptance rate. Furthermore, as the length of the SFC request grows, the performance differences between the RB-DSD and RG-DSD schemes and the proposed BR-DSD scheme become pronounced. Specifically, the RB-DSD scheme attempts to balance CPU resources consumption across more nodes, requiring extra bandwidth and CPU resources to facilitate communication, which ultimately leads to a decline due to resource scarcity. On the other hand, the RG-DSD scheme concentrates resource demands on a single node in an attempt to conserve overall network resources, which makes it more susceptible to network congestion and ultimately leads to a decline in the service acceptance rate. Consequently, the superiority of the BR-DVD algorithm becomes increasingly evident as the length of the SFC request escalates.

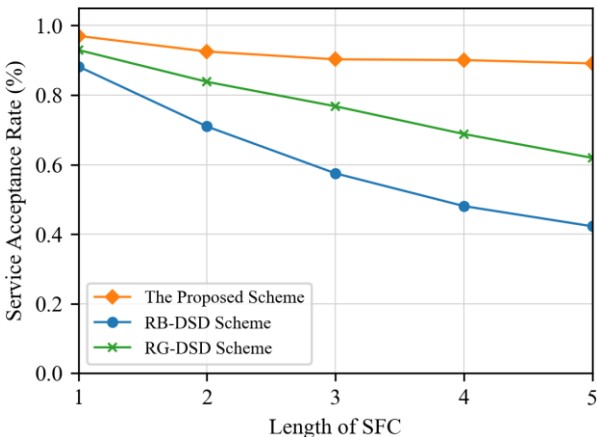

Fig. 6. The performance of service acceptance rate versus the length of SFC.

Fig. 7 and Fig 8 illustrate how the service acceptance rate varies with the capacity of bandwidth and CPU resources. As the capacity of either bandwidth or CPU resources increases, the service acceptance rates of all three schemes initially rise and then stabilize at a level less than 1, indicating that both types of resources impact the service acceptance rate. Notably, as bandwidth resources increase, the rise in the service acceptance rate is more significant, suggesting that in our network, the service acceptance rate is more influenced by bandwidth resources than by CPU resources.

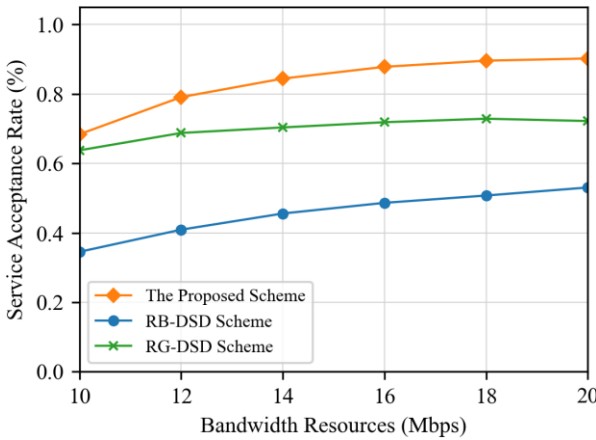

Fig. 7.   The service acceptance rate varies with the bandwidth resources.

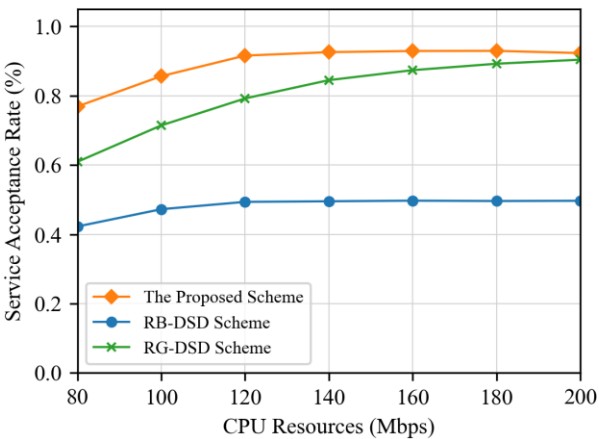

Fig. 8.   The service acceptance rate varies with the CPU resources.

## V. CONCLUSION

In this paper, we propose the dynamic SFC deployment optimization problem in an NFV-enabled offshore edge computing network. We jointly optimize the VNF deployment and routing with the objective of maximizing the service acceptance rate. Due to the NP-hard, we proposed the BR-DSD scheme to solve the optimization problem. The BR-DSD scheme considers both network resource availability and betweenness centrality when allocating resources for deploying

SFCs. To be specific, for each service request, we employ the DLDP algorithm to find routing paths. Based on these paths, use the BRVD algorithm to determine the VNF placement strategy. Simulation results demonstrate that the proposed BR-DSD scheme outperforms the baseline schemes and achieves high performance in terms of the service acceptance rate.

### ACKNOWLEDGMENT

The work was supported by the National Natural Science Foundation of China (No. 62371085, No. 51939001) and Fundamental Research Funds for the Central Universities (No. 3132023514).

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
