# OpenReview forum: "Dynamic Service Function Chain Deployment in NFV-enabled Offshore Edge Computing Networks"
_IEEE.org/ICIST/2024/Conference — IEEE ICIST 2024 Conference Submission_

### Official Review · Reviewer_VsiT · 2024-08-23
**Generally good, but there are still some problems.**

**Rating:** 8
**Confidence:** 4

**Review:**

The manuscript titled "Dynamic Service Function Chain Deployment in NFV-enabled Offshore Edge Computing Networks" addresses the challenge of deploying service function chains (SFCs) in offshore edge computing environments, which are enabled by Network Function Virtualization (NFV).  The authors formulate the dynamic SFC deployment optimization (DSDO) problem, focusing on maximizing the service acceptance rate by jointly optimizing the deployment of virtual network functions (VNFs) and routing.

Comments:

1.The manuscript introduces a novel approach by integrating betweenness centrality with resource availability for optimizing SFC deployment in offshore edge networks. However, the paper would benefit from a clearer comparison with existing methods to better highlight the unique advantages of the BR-DSD scheme.

2.While the manuscript is well-organized, some of the technical sections, particularly those describing the algorithms, could be made clearer. Simplifying these explanations or including additional diagrams could enhance reader comprehension.

3.The experimental results are convincing, but expanding the range of test scenarios, such as varying levels of network congestion or different service request patterns, would provide a more robust evaluation of the BR-DSD scheme's performance.

---

### Official Review · Reviewer_GRYE · 2024-08-24
**Review Comments for Manuscript No. 181**

**Rating:** 8
**Confidence:** 4

**Review:**

1. There are some noticeable formatting errors in the manuscript. The authors should carefully review and ensure that the paper meets the IEEE conference paper standards.

2. The text size in Figure 1 is inconsistent with that in other figures and should be adjusted. Additionally, the references need further checking to ensure accuracy and consistency.

3. The Dynamic Service Function Chain (SFC) Deployment Optimization (DSDO) problem proposed by the authors—how does it differ from and what are the challenges compared to other studies in NFV-supported offshore edge computing networks? The authors should provide more detailed explanations to highlight these contrasts.

4. In modeling the DSDO problem, has the manuscript fully considered various constraints present in real-world network environments, such as latency and data throughput limitations? Are the model's assumptions reasonable and generalizable? Is the design of the BR-DSD approach, which simultaneously considers network resources and betweenness centrality, appropriate?

5. Has the implementation of the BR-DSD approach taken into account the complexity and scalability of the algorithm? Were potential challenges in practical deployment, such as real-time requirements and dynamic changes in network topology, considered?

---

### Official Review · Reviewer_mZ4G · 2024-08-24
**This article is very interesting and a good one.**

**Rating:** 7
**Confidence:** 5

**Review:**

1. The motivations should be further highlighted in the manuscript, e.g., what problems did the previous works exist? How to solve these problems? The authors may consider analyzing the problems of the previous works and how to address these problems with the proposed method.
2. The literature review in the Introduction surveys some previous work on the solutions and methods to address the SFC deployment problem, while comparation among these methods which is the most important part is neglected.
3. The quality of language needs significant improvement, and professional editing may be necessary.

---

### Decision · Program_Chairs · 2024-09-08

Accept (Oral)